# Effects of the Three-Direction Movement Control Focus Complex Pain Program and Neurodynamic Focus Complex Pain Program on Pain, Mechanosensitivity, and Body Function in Taekwondo Athletes with Non-Specific Low Back Pain: A Preliminary Study

**DOI:** 10.3390/healthcare12040422

**Published:** 2024-02-06

**Authors:** Hong-gil Kim, Ju-hyeon Jung, Song-ui Bae

**Affiliations:** 1Department of Physical Therapy, Graduate School, Dong-Eui University, Busan 47340, Republic of Korea; rlaghdrlf456@gmail.com (H.-g.K.); song2732@naver.com (S.-u.B.); 2Department of Physical Therapy, College of Nursing, Healthcare Sciences and Human Ecology, Dong-Eui University, Busan 47340, Republic of Korea

**Keywords:** non-specific low back pain, Taekwondo athletes, movement control exercise, neurodynamic technique

## Abstract

We aimed to determine the effects of three-direction movement control focus complex pain program (3D-MCE) and neurodynamic focus complex pain program (NDT) on pain, mechanosensitivity, and body function in Taekwondo athletes with non-specific low back pain. This study used a two-group pretest–posttest design and was conducted at a university physiotherapy lab and training center. It included 21 Taekwondo athletes with non-specific low back pain from a Taekwondo studio and a University in Busan. Participants were divided into a 3D-MCE group (n = 10) and an NDT group (n = 10). The numerical rating pain scale (NRPS), pain pressure threshold (PPT), movement analysis, and Oswestry Disability Index (ODI) were measured before and after the intervention. The intervention was performed for 45 min twice a week for 4 weeks. Each group performed movement control exercises and neurodynamic techniques. The NRPS, motion analysis, and ODI were significantly changed after the intervention in the 3-DMCE group. The NRPS, PPT, and ODI changed significantly after the intervention in the NDT group. Moreover, the PPT and motion analysis showed significant differences between the two groups. For Taekwondo athletes with non-specific low back pain, 3D-MCE improved the stability control ability of the lumbar spine. It was confirmed that neurodynamic techniques reduce muscle and nerve mechanosensitivity.

## 1. Introduction

Lumbar pain develops into chronic or recurrent pain with increased duration and repeated intervals of pain as a result of combined factors, such as physical defects, psychological aspects, lifestyle, stress, and job-related issues [1]. As Taekwondo became a sport type that involves matches and competitions from the previous demonstration type, athletes are required to perform higher levels of techniques, which leads to lumbar pain and various injuries [2]. Previous studies investigating the injuries related to Taekwondo claim that the most common injuries are those of peripheral nerves and movement control disabilities regarding the flexion of the lumbar bone in Taekwondo athletes [3,4,5]. This is because Taekwondo athletes must repeat the movements in the direction of lumbar flexion for the kicking motion of a high-level difficulty [6]. The repeated lumbar stress through flexion causes a movement control disability which increases micro-injuries of the lumbar tissues [7]. As a result, unnecessary contraction and tension of the lumbar muscles and the sensitization of surrounding muscles lead to pain [8]. In addition to this mechanism of pain, lumbar pain is accompanied by circulatory disorders and edema of the peripheral nerves themselves, and symptoms of involuntary muscle contraction [5,9]. Thus, Taekwondo athletes frequently experience lumbar pain due to movement disabilities and problems with nerve mobility as they continue to perform repeated movements [10]. Recent studies reported that athletes with injuries opt to undergo surgical, drug, and physical therapy to treat such symptoms, spending much time and cost to alleviate lumbar pain [11]. Earlier, ultrasound, electrical stimulation, therapeutic taping, and manual therapy were widely used to control pain in athletes with lumbar pain, while the athletes were recommended to take a break from training to give time for the pain to subside [12]. However, recent studies on rehabilitation for lumbar pain reported that conventional methods only allow transient control of pain as they rely on pain control via the stimulation of peripheral receptors [13]. This is attributable to the treatments having been focused on the tissue level, muscles, joints, and ligaments, without taking into consideration the various causes of pain among the athletes [14].

Meanwhile, previous studies suggested that the movement control exercise (MCE) and neurodynamic technique (NDT) can be effective interventions for patients with non-specific low back pain [15,16]. The effects of MCE included the reduction of unnecessary mechanical loading induced by repeated stress on the tissues and of the sensitivity of peripheral receptors [9,17]. Additionally, NDT has been recommended as a method to improve reduced peripheral nerve mobility [6]. The result can be the alleviation of pain as the movement of nerve tissue is enhanced and the pressure on the nerve tissue is reduced, leading to improved circulatory capacity [18,19]. In many studies on non-specific low back pain patients, MCE and NDT were reported to be effective in controlling low back pain [19]. However, in patients with non-specific low back pain, the effect of three-direction movement control exercise (3D-MCE) on low back pain control has not been verified. In addition, there is a lack of studies confirming which elements of the pain mechanisms were improved by the particular intervention in the previous study. Moreover, few studies have investigated the effects of 3D-MCE and NDT on pain, mechanosensitivity, and physical function in Taekwondo athletes with non-specific low back pain.

This study aimed to compare the effects of each complex pain program that emphasized movement control and peripheral nerve neurodynamics on Taekwondo players complaining of non-specific low back pain. Additionally, we sought to identify pain mechanisms that frequently occur in Taekwondo players and propose a pain management program appropriate for the pain mechanisms.

## 2. Materials and Methods

### 2.1. Study Design

This study was approved by the Institutional Review Board at Dong-Eui University (IRB No. DIRB-202202-HR-R-02), and informed consent was obtained from participants.

This quasi-experimental study recruited 21 Taekwondo athletes with lumbar pain. However, one person was excluded because he received other medical treatment due to acute pain during the intervention and did not meet the inclusion criteria. This study considered the homogeneity of the two groups. One subject was excluded, and, finally, 20 athletes participated in the experiment.

The participants in this study were 20 individuals who voluntarily agreed to participate after receiving a detailed explanation of the study protocol. Informed consent was obtained from all subjects involved in the study. All subjects provide informed consent in the study. The 3-direction movement control focus complex pain program (3D-MCE) group and neurodynamic focus complex pain program (NDT) groups were divided via random sampling subjects who satisfied the participants’ inclusion criteria. Then, 16 cards, marked “A” for the intervention group, and another 16 cards, marked ‘B’ for the control group, were placed in an opaque envelope and drawn by the participants. This investigator was not involved in the intervention and assigned the participants to random groups.

To verify the effectiveness of each intervention, the lumbar pain, mechanosensitivity, and body function were measured before and after the intervention. The pre- and post-evaluations were conducted blindly so that evaluators different from people who intervened could not confirm the group to which the subject belonged (Figure 1). The program of intervention was 45 min per session and two sessions a week for 4 weeks [20]. For the 3D-MCE group, the 3D-MCE of lumbar flexion, extension, and rotation was combined with transcutaneous electrical nerve stimulation (TENS) therapy and therapeutic massage of the lumbar muscles. For the NDT group, the neurodynamic sliding technique on sciatic nerves was combined with TENS therapy and therapeutic massage of lumbar muscles. Each intervention program was applied to each participant by a physical therapist with more than six years of experience, and feedback and corrections were provided if there were errors in the subject’s movements when applying the intervention.

While applying the intervention, the subject’s exercise intensity was maintained at a grade of 4 to 5 (somewhat hard) by checking Ratings of Perceived Exertion (RPE), and sufficient rest was taken if the subject complained of discomfort. The intervention effects were determined based on the pretest of numerical rating pain scale (NRPS), mechanosensitivity, movement analysis (MA), and Oswestry Disability Index (ODI) before intervention and the posttest of the same instruments 4 weeks after intervention. The flowchart of the study design and methods is given in Figure 1.

### 2.2. Participants

The inclusion criteria, formed according to previous studies [15,21], were as follows: two or more times of lumbar pain experienced in the past 6 months; difficulty in athletic performance due to pain; ODI ≥ 6; NRPS ≥ 4; and a positive result on the backward push test (BPT) with pain during BPT and collapsed position at the lumbar area before 30° hip flex. The athletes satisfying these inclusion criteria were selected through random sampling and were assigned to 3D-MCE and NDT groups.

The exclusion criteria were as follows: pain experienced due to an internal organ disease such as cancer or cardiac disease in addition to lumbar pain; a risk of increased pain during intervention due to an already high intensity of pain; a history of surgery, scoliosis, or neurological symptoms; and skin sensitivity that prevents measurements and interventions.

For 10 participants (5 participants each in the intervention and control groups), the preliminary sample size estimated by the G*power program (ver. 3.1.9.2, Heinrich Heine University, Düsseldorf, Germany) under the conditions of effect size being 1.3, significance level of 0.05, and testing power of 0.8 was 18. Based on this, 20 individuals were recruited in consideration of a 10% drop-out rate.

### 2.3. Outcome Measures

#### 2.3.1. Pain

The general level of lumbar pain felt by the participating Taekwondo player was measured using NRPS [22].

#### 2.3.2. Mechanosensitivity

To measure the mechanosensitivity, an Algometer (Pain Test™ FPX 25 Algometer; Wagner Instruments, Greenwich, CT, USA) was used for checking the pressure pain threshold (PPT), and the pressure at the first sensation of pain was measured. To ensure that the mechanosensitivity of the pain area was measured, the area of measurement was a muscle belly area at the erector spinae, 2 cm adjacent to the third lumbar spinous process, to which pressure was applied [23]. To determine the nerve mechanosensitivity, pressure was applied to the tibial nerve at the boundary of the popliteus [24]. The participant was guided to raise a hand when they sensed the pressure as pain. The measurements were taken three times, with 30 s rest between each measurement [25].

#### 2.3.3. Motion Analysis

To analyze the lumbar movements, a portable automatic analyzer (Myomotion; Noraxon, Scottsdale, AZ, USA) was used upon BPT. The sensors were attached to the upper thoracic (T1, 2) and lower thoracic (T11, 12) areas to measure the thoracic flexion and to the lower thoracic (T11, 12) and pelvic (base of the sacrum at posterior superior iliac spine (PSIS)) areas to measure the lumbar flexion. Additional sensors were attached to the thigh bone and pelvic (base of the sacrum at PSIS) areas to measure range of motion (ROM) changes during hip joint flexion. The BPT was repeated three times for 3 s [26,27]. The start position for the test was 90° flexion of the knee and shoulder joints. The end position was a 30° flexion of the hip joint with the curves of the lumbar and spinal bones maintained during the BPT. The mean of triplicate measurements of the level of lumbar flexion within the range of 30° hip joint flexion from the start position was used in the analysis [28].

#### 2.3.4. Oswestry Disability Index

Participants completed a questionnaire before and after the intervention. The ODI questionnaire consists of ten items related to daily activities to assess the following factors: pain intensity, self-management, sitting, standing, sexual life, lifting, social life, sleep, and travel. Each item is a yes–no question regarding the state in the past 24 h, and it is rated on a scale of 0 to 5. The Korean version excludes the item of sexual life with a total score of 45. Higher scores indicate higher disorder levels [16].

### 2.4. Interventions

#### 2.4.1. 3-Direction Movement Control Exercise

The Taekwondo athletes in the 3D-MCE group performed the MCE in three directions: lumbar flexion, extension, and rotation. The athletes performed the exercise up to the maximum achievable level from the start position until they felt pain or impossible to maintain the posture during the exercise. The 3D-MCE was performed in 3 sets of 15 repetitions. The athletes were trained to perform the exercise in the direction of flexion up to 50° hip joint flexion while maintaining the curve of the lumbar bones in a standing posture. For the exercise in the direction of extension, pressure feedback equipment (Stabilizer, Chattanooga, TN, USA) was placed at the center of the abdomen of the player in a prone position. The initial pressure was set to 70 mmHg, and the player was trained to adjust the pressure to 60 mmHg by contracting the abdominal muscles, and performing the extension up to 0° hip joint while maintaining the pressure of the equipment at the center of the abdomen at 60 mmHg. For the exercise in the direction of rotation, the pressure feedback equipment was placed under the PSIS on the antagonistic side prior to leg lift by the player in an identical posture as with the extension. The initial pressure was set to 70 mmHg and the player was trained to maintain the pressure at 70 mmHg without a rise or fall in the leg lift to 0° hip joint [29,30] (Figure 2, Table 1).

#### 2.4.2. Neurodynamic Technique

The NDT performed by the NDT group in this study was based on a sliding technique known to be effective in enhancing nerve mobility and reducing peripheral nerve edema and inflammation through the movements of the body [9,31]. The Taekwondo athletes in a slump position were guided not to exhibit a joint movement other than the knee and neck flexion and extension to minimize the impact on other joints. In addition, the athletes were guided to perform neck extension with knee extension and knee flexion with neck flexion to avoid nerve traction or pain in NDT [32]. The NDT was applied for 15 min on each leg 15 times per set for 3 sets [9,31] (Figure 3, Table 1).

#### 2.4.3. Therapeutic Massage

In this study, a therapeutic massage using the deep friction technique was performed by a physical therapist to reduce the tissue mechanosensitivity and enhance flexibility. The massage was applied to the hamstring and erector spinae muscles on both sides of each Taekwondo player. The massage was performed for 15 min, was stopped when the player complained of pain, and then resumed after adequate rest [33] (Table 1).

#### 2.4.4. Transcutaneous Electrical Nerve Stimulation Therapy

The TENS (Saehan LT1061, Supia, Republic of Korea) was applied to the Taekwondo athletes in a prone position with the electrodes placed at the area of lumbar pain and the respective peripheral nerve pathways toward the lower limbs. The pulse duration and exercise intensity were ≤150 μs and the frequency was 80 Hz. The intensity was decreased upon skin irritation or pain, and the therapy was resumed after adequate rest [34] (Table 1).

### 2.5. Statistical Analysis

The data collected in this study were analyzed using the SPSS 22.0 (IBM Corp., Amonk, NY, USA) for Windows, and the level of significance for statistical validation was set at 0.05. The participants’ general characteristics were analyzed through descriptive statistics, and a Mann–Whitney U test was performed to test the inter-group homogeneity. The Shapiro–Wilk test was used to check normal distribution. As all variables were not shown to satisfy normality, a non-parametric analysis method was used.

A Mann–Whitney U test was performed to analyze the groups’ pretest and posttest changes (mean difference value) in the inter-group. The Wilcoxon signed rank test was conducted to examine the intra-group variation.

## 3. Results

### 3.1. Characteristics of Participants

Twenty Taekwondo athletes with non-specific low back pain participated in this study, and the general characteristics of the participants and the homogeneity test result of the two groups in this study are as follows (Table 2).

### 3.2. Numerical Rating Pain Scale

Significant changes in pain were observed in both 3D-MCE and NDT groups after intervention (*p* < 0.05). The between-group comparison indicated no significant difference in NRPS (*p* > 0.05) (Figure 4, Table 3).

### 3.3. Pressure Pain Threshold

After the intervention, the 3D-MCE group showed significant changes in the mechanosensitive nerves and the erector spinae on the left side (*p* < 0.05). But, the 3D-MCE group showed no significant changes in the mechanosensitive of the erector spinae on the right side (*p* > 0.05). In contrast, after intervention, the NDT group showed significant changes across all muscles and nerves (*p* < 0.05).

The between-group comparison indicated no significant difference in muscle mechanosensitivity for the erector spinae on the right side (*p* > 0.05). In contrast, the erector spinae on the left side had a significant difference (*p* < 0.05). The nerve mechanosensitivity was a significant difference across all tested areas (*p* < 0.05) (Figure 4, Table 4).

### 3.4. Motion Analysis

After the intervention, the 3D-MCE group showed significant changes in the lumbar flexion angle (*p* < 0.05), while the NDT group showed no significant change in the lumbar flexion angle (*p* > 0.05). The between-group comparison indicated a significant difference in lumbar flexion angle (*p* < 0.05) (Figure 4, Table 5).

### 3.5. Oswestry Disability Index

Both 3D-MCE and NDT groups showed significant changes in the ODI after intervention (*p* < 0.05). In contrast, the between-group comparison indicated no significant difference in ODI (*p* > 0.05) (Figure 4, Table 5).

## 4. Discussion

This study aimed to verify the effects of a 4-week intervention of 3D-MCE and NDT on pain, mechanosensitivity, and body functions in Taekwondo athletes with lumbar pain and to identify the pain mechanism frequently arising in Taekwondo athletes to suggest a suitable intervention for the pain mechanism.

In previous studies, a numerical pain scale was used to measure the changes in lumbar pain [35,36]. In this study, the results revealed significant pretest–posttest differences in pain between and within the two groups. This coincided with previous studies reporting the positive effects of 3D-MCE and NDT on pain control in Taekwondo athletes with lumbar pain [37,38]. Regarding 3D-MCE, previous studies reported the improvement of pain and the injuries of adjacent tissues as the neutral position of the spine was perceived and excessive joint movements in the vicinity were controlled [7,8]. Regarding NDT, previous studies reported that the mobility of nerves leading from the lumbar to the lower limb areas was improved [19]. Based on the results, NDT was shown to be effective for pain control to restore the normal state of nerves from edema, inflammation, and hypoxia, thereby reducing the sensitivity of the sensory receptors [9,19]. These studies lend support to the pain improvement in the two groups of this study.

Mechanosensitivity is concerned with the sensitive state of nerve activation upon pressure or tension [18]. The mechanosensitivity at the lumbar area increased due to repeated stress, implying that pain can be sensed at a greater intensity even with low stimulations [39]. As a way to measure mechanosensitivity, previous studies suggest the measurement of pressure pain threshold (PPT) [27,36].

In this study, PPT was measured for the erector spinae and tibial nerve to examine the mechanosensitivity of the muscles in the lumbar area and peripheral nerves from the lumbar bones, respectively [24,25].

The result was interpreted such that an increase in PPT indicated a fall in mechanosensitivity [40]. In previous studies, mechanosensitivity was shown to decrease through 3D-MCE and NDT [29,41].

Previous studies reported that 3D-MCE contributed to improving the muscle mobility and cognitive abilities of the body while being effective in reducing muscle mechanosensitivity [29]. However, in this study’s results, the 3D-MCE group did not show significant changes in the right side of the erector spinae at the pre-post test. However, it showed significant changes in the left side of the erector spinae. This result is because 3D-MCE intervention did not consistently influence reducing muscle mechanosensitivity. On the other hand, in the NDT group, the PPT test results on the right and left erector spinae showed uniform pre-post changes of more than 1 lb/cm. This result is related to the fact that the left erector spinae showed no significant difference in comparing the two groups. This is because the 3D-MCE focused on reducing the stress in the surrounding tissues through normal movements rather than directly affecting mechanosensitivity. On the other hand, the neurodynamic technique applied to the NDT group has produced more consistent results than movement control exercises by improving the mobility and circulation of peripheral nerves directly connected to muscles and directly affecting the sensory areas of nerves and muscle tissue.

In addition, although there is no statistical difference in the pretest results in the 3D-MCE group compared to other groups, it can be confirmed that the average sensitivity value of the left muscle is somewhat lower than the result of the right muscle. These differences in pretest values affected the statistical values before and after the intervention and the consequences of comparison between groups.

An important point that can be seen from these results is that Taekwondo players who complain of non-specific low back pain may have a left–right imbalance in the mechanosensitivity of the erector spinal muscles, which is influenced by the degree of use, position of the dominant foot, training method, and how the athlete utilizes the skill. Future research needs to identify intervention methods to improve left–right imbalance in Taekwondo players who exhibit mechanical sensory imbalance between left and right muscles.

Furthermore, previous studies reported that NDT enhanced the blood flow for nerves and surrounding tissues to reduce mechanosensitivity [41]. In line with this, the NDT group showed significant changes across all tested areas.

The between-group comparison on mechanosensitivity in this study revealed a significant difference between the NDT and 3D-MCE groups. This is consistent with a previous report on the positive effects of NDT in reducing muscle and nerve mechanosensitivity by reducing the paresthesia and sensitivity of sympathetic nerves and improving the nociceptors [42]. It was thus shown that NDT was an intervention to reduce mechanosensitivity by enhancing the blood flow to the nerves and muscles and reducing the senses and tissue hypersensitivity.

In previous studies, motion analysis was used to examine the movement disability of lumbar bones [28,38]. Based on the BPT used in previous studies, a motion analysis of lumbar bones was performed in this study to verify the improvement of the movement disability of lumbar bones in Taekwondo athletes [43]. An increase in the lumbar flexion angle upon BPT in this study indicates dysregulation in the direction of lumbar flexion. Moreover, excessive lumbar mobility without the ability to maintain the curve of the lumbar bones during BPT indicates a high probability of increased stress on the surrounding tissues [29,44,45]. Hence, the decrease in lumbar flexion angle in this study indicates improved stability of lumbar bones in motion. The results of this study revealed that the lumbar flexion angle decreased significantly after intervention in the 3D-MCE group, whereas the NDT group showed no significant change after intervention. The between-group comparison further demonstrated a significant difference as the level of variation was higher in the 3D-MCE group than in the NDT group. This suggested that 3D-MCE was an intervention to enhance the stability of lumbar bones. In a previous study, MCE was shown to disperse the loading on each segment of the body through the perception of the normal alignment and isolated motion of each segment [38]. In addition, it was shown that the retraining of the movements to increase the agonistic and antagonistic muscle coordination was effective for pain control [38,46]. The intervention given to the MCE group in this study likewise applied the exercise in the directions of lumbar flexion, extension, and rotation to enhance the agonistic and antagonistic muscle coordination, and this improved the motion analysis result in the 3D-MCE group.

In contrast, the lack of significant change in motion analysis results in the NDT group is likely to be because the NDT is an intervention focused on mechanosensitivity rather than the improvement of cognitive abilities of the segments of the body.

The ODI has been widely used across a number of previous studies to objectively measure the intensity of pain in Taekwondo athletes with recurrent lumbar pain and to obtain numerical values of the physical functional levels related to dysfunctions and cognitive impairments [46,47,48].

In this study, the ODI varied significantly between pre- and post-intervention measurements in the two groups, while no significant between-group difference was observed. The results suggested that both 3D-MCE and NDT were effective interventions to improve the daily activities of Taekwondo athletes with lumbar pain. This is presumed to be because the two intervention methods ultimately led to the improvement of pain and the reduced pain could resolve the limitations on body functions to improve the movements.

An interesting point in the ODI result of this study is that, when the scores of lifting, sitting, and standing that are closely associated with motion were compared between the two groups, the scores were lower by 6% in the 3D-MCE group and by 3.8% in the NDT group. This confirmed that the 3D-MCE was effective in improving the factors of ODI related to movements. The analysis was in line with the previous study reporting the improved abilities of daily activities through the improvement of lumbar control abilities through 3D-MCE [46].

The results of this study may be summarized as follows: In Taekwondo athletes with lumbar pain, the repeated kicking motion increases the load on the lumbar area and, as a result, the sensitivity of peripheral nerves increases and dysregulation of motion occurs. Meanwhile, the 3D-MCE and NDT in this study were effective in improving pain and body functions in Taekwondo athletes with lumbar pain. Moreover, the 3D-MCE is effective for improving the lumbar control abilities with consequent functional improvement regarding motion, while the NDT is effective for reducing tissue mechanosensitivity. This could be attributed to the fact that the methods focused on different factors to alleviate the cause of the pain mechanism. Additionally, the ability to differentiate the specific area of the problem causing the lumbar pain was shown to be critical in clinical settings for applying the interventions to Taekwondo athletes with lumbar pain.

Recent studies have shown that various types of pain, including lumbar pain, occur through five mechanisms: increased sensitivity of peripheral sensory receptors, damage of peripheral nerves, central sensitization, reduced control of motion, and psychosocial factors [49]. This study verified the effectiveness of neurodynamic and movement control focus complex interventions. Additionally, in previous studies, the pain mechanism is recommended to apply a combined treatment to ensure adequate pain control [31,50,51]. Therefore, based on the results of this study, the effect of the combination of a new complex intervention program on pain control should be verified, and follow-up research should be conducted to develop intervention programs, including central sensitization and psychosocial factors.

Limitations of this study include the insufficient number of subjects who participated in the survey, the results of the continuous effect of the intervention, and the recurrence of subjects’ LBP after the intervention, which could not be confirmed. These limitations should be resolved for the effects of 3D-MCE and NDT focus pain control intervention to be verified in sports athletes with lumbar pain and various other musculoskeletal pains.

## 5. Conclusions

The 3D-MCE and NDT applied to Taekwondo athletes with lumbar pain effectively improve pain, mechanosensitivity, and body functions. The 3D-MCE results in a higher level of improvement in the lumbar control abilities, and the NDT group results in a higher level of reduction in mechanosensitivity in the nerve and muscle tissues. Furthermore, as the cause of pain varies according to the pain mechanism, a suitable and specific intervention should be applied.

## Figures and Tables

**Figure 1 healthcare-12-00422-f001:**
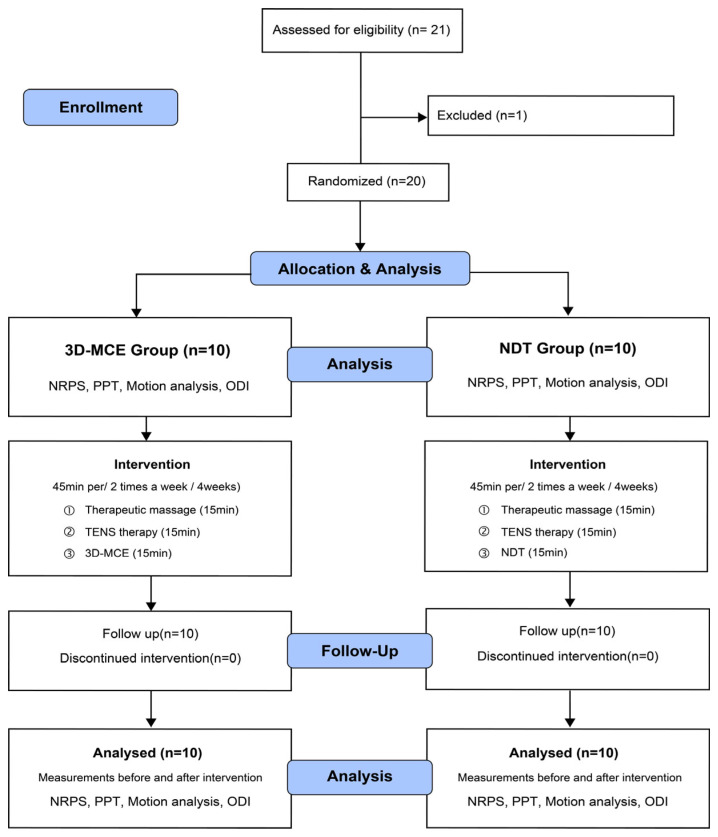
Flowchart of the study design and methods. 3D-MCE: 3-direction movement control focus complex pain program; NDT: neurodynamic focus complex pain program; PPT: pressure pain threshold; TENS: transcutaneous electrical nerve stimulation; NRPS: numerical rating pain scale; ODI: Oswestry disability index.

**Figure 2 healthcare-12-00422-f002:**
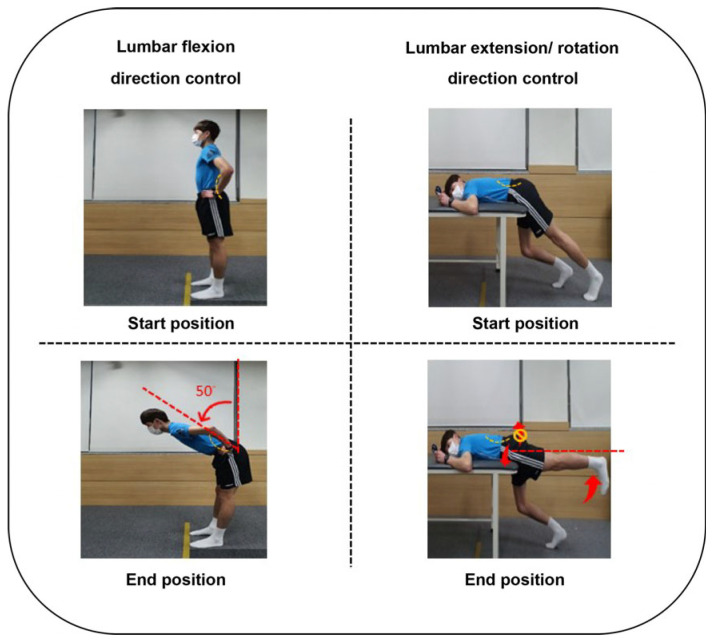
3-direction movement control exercise program.

**Figure 3 healthcare-12-00422-f003:**
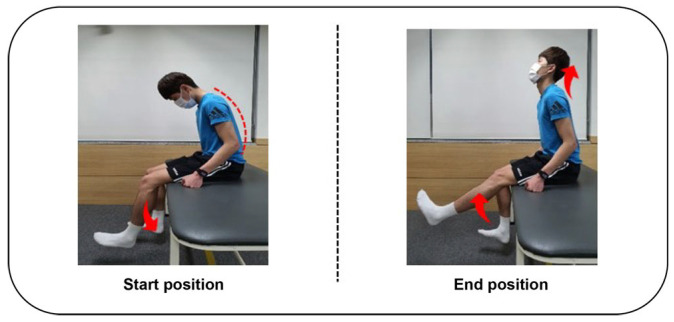
Neurodynamic sliding technique.

**Figure 4 healthcare-12-00422-f004:**
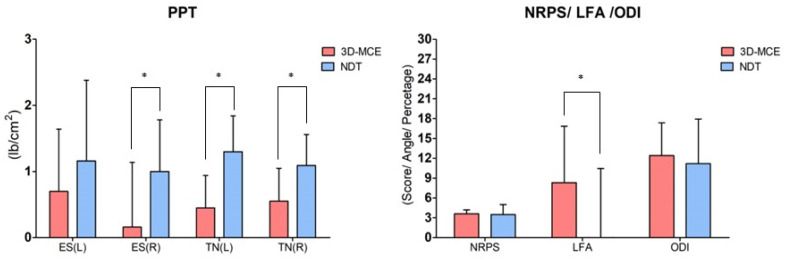
Comparison results for each measurement factor (mean difference value) between the two groups. Values are means ± SD. * Significant difference between groups (*p* < 0.05). 3D-MCE: 3-direction movement control focus complex pain program; NDT: neurodynamic focus complex pain program; NRPS: numeral rate pain scale; PPT: pressure pain threshold; ES: erector spinae; TN: tibial nerve; LFA: lumbar fexion angle; ODI: Oswestry disability index.

**Table 1 healthcare-12-00422-t001:** The intervention program by group.

3D-MCE Group	NDT Group
1. Therapeutic massage (15 min) Hamstring Erector spinae	1. Therapeutic massage (15 min) Hamstring Erector spinae
2. TENS therapy (15 min) Lumbar Lower extremity nerve root	2. TENS therapy (15 min) Lumbar Lower extremity nerve root
3. Movement control exercise (15 min) MCE of lumbar flexion MCE of lumbar extension MCE of lumbar rotation	3. Neurodynamic technique (15 min) Levator scapulae Upper trapezius

3D-MCE: 3-direction movement control focus complex pain program; NDT: neurodynamic focus complex pain program; TENS: transcutaneous electrical nerve stimulation therapy.

**Table 2 healthcare-12-00422-t002:** Demographic and anthropometric characteristics of the subjects and homogeneity test of variables between the two groups (n = 20).

Variables			3D-MCE (n = 10)	NDT (n = 10)	*Z*	*p*
Age (years)			17.30 ± 3.30	20.40 ± 3.77	−1.95	0.719
Height (cm)			168.70 ± 12.33	168.1 ± 6.90	0.134	0.011
Body mass (kg)			64.00 ± 20.23	62.90 ± 14.43	0.140	0.296
NRPS (Score)			6.4 ± 1.34	6.6 ± 0.96	−0.381	0.226
PPT (lb/cm)	ES	Lt	3.07 ± 0.28	3.32 ± 0.39	−0.509	0.256
		Rt	3.32 ± 0.25	3.21 ± 0.26	0.097	0.945
	TN	Lt	2.43 ± 0.54	2.80 ± 0.93	−1.86	0.065
		Rt	2.92 ± 0.87	2.48 ± 0.69	1.23	0.994
LFA (°)			21.27 ± 7.41	15.67 ± 10.02	1.64	0.216
ODI (%)			9.3 ± 3.91	8.6 ± 3.37	0.428	0.584

Mean ± standard deviation; 3D-MCE: 3-direction movement control focus complex pain program; NDT: neurodynamic focus complex pain program; NRPS: numeral rate pain scale; PPT: pressure pain threshold; ES: erector spinae; TN: tibial nerve; LFA: lumbar fexion angle; ODI: Oswestry Disability Index.

**Table 3 healthcare-12-00422-t003:** Pre and post-test outcomes in NRPS measures in 3D-MCE and NDT before and after treatment.

	Group	Pretest	Posttest	Z	*p*	Mean Difference	Z	*p*
		M ± SD				M ± SD		
NRPS (Score)	3D-MCE	6.4 ± 1.34	2.8 ± 1.31	−2.85	0.004 *	3.6 ± 0.60	−0.351	0.725
	NDT	6.6 ± 0.96	3.1 ± 0.73	−2.82	0.005 *	3.5 ± 1.50		

* *p* < 0.05, mean ± standard deviation; 3D-MCE: 3-direction movement control focus complex pain program; NDT: neurodynamic focus complex pain program; NRPS: numeral rate pain scale.

**Table 4 healthcare-12-00422-t004:** Pre and post-test outcomes in PPT measures in 3D-MCE and NDT before and after treatment.

		Group	Pretest	Posttest	Z	*p*	Mean Difference	Z	*p*
			M ± SD	M ± SD			M ± SD		
PPT (lb/cm)	ES(L)	3D-MCE	3.07 ± 0.89	3.77 ± 0.97	−2.09	0.037 *	0.70 ± 0.0.94	−0.794	0.427
		NDT	3.32 ± 1.26	4.49 ± 1.08	−2.70	0.007 *	1.16 ± 1.22		
	ES(R)	3D-MCE	3.25 ± 0.80	3.41 ± 1.03	−0.663	0.507	0.16 ± 0.0.98	−2.04	0.041 *
		NDT	3.21 ± 0.84	4.22 ± 0.91	−2.80	0.005 *	1.00 ± 0.73		
	TN(L)	3D-MCE	2.43 ± 0.54	2.88 ± 0.49	−2.34	0.019 *	0.45 ± 0.0.49	−2.79	0.005 *
		NDT	2.80 ± 0.93	4.11 ± 0.69	−2.80	0.005 *	1.30 ± 0.54		
	TN(R)	3D-MCE	2.92 ± 0.87	3.47 ± 0.97	−2.49	0.009 *	0.55 ± 0.0.50	−2.00	0.045 *
		NDT	2.48 ± 0.69	3.58 ± 0.77	−2.80	0.005 *	1.09 ± 0.047		

* *p* < 0.05, mean ± standard deviation; 3D-MCE: 3-direction movement control focus complex pain program; NDT: neurodynamic focus complex pain program; PPT: pressure pain threshold; ES: erector spinae; TN: tibial nerve.

**Table 5 healthcare-12-00422-t005:** Pre and post-test outcomes in LFA and ODI measures in 3D-MCE and NDT before and after treatment.

	Group	Pretest	Posttest	Z	*p*	Mean Difference	Z	*p*
		M ± SD	M ± SD			M ± SD		
LFA (°)	3D-MCE	21.27 ± 7.41	12.94 ± 5.83	−2.39	0.017 *	8.33 ± 8.51	−1.96	0.049 *
	NDT	15.67 ± 10.02	15.65 ± 9.49	−0.051	0.959	0.02 ± 10.41		
ODI (%)	3D-MCE	18.6 ± 7.82	6.2 ± 4.04	−2.81	0.005 *	12.4 ± 4.96	−0.687	0.492
	NDT	17.2 ± 6.74	6 ± 1.88	−2.81	0.005 *	11.2 ± 6.74		

* *p* < 0.05, mean ± standard deviation; 3D-MCE: 3-direction movement control focus complex pain program; NDT: neurodynamic focus complex pain program; LFA: lumbar fexion angle; ODI: Oswestry disability index.

## Data Availability

The data presented in this study are available on request from the corresponding author.

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
