# Peer review of "Effects of the Three-Direction Movement Control Focus Complex Pain Program and Neurodynamic Focus Complex Pain Program on Pain, Mechanosensitivity, and Body Function in Taekwondo Athletes with Non-Specific Low Back Pain: A Preliminary Study"

_healthcare, 2024, doi:10.3390/healthcare12040422_

Round 1

Reviewer 1 Report

Comments and Suggestions for Authors

Introduction

Several studies have demonstrated that both MCE and NDT are effective for individuals with NSLBP. The directions of the movement control are flexion, extension and rotation. In the introduction the research question needs to be clearly stated as well as the novelty of the study.

Methods: Please clarify the selection of the specific protocol. Why have you chosen a short-term intervention (4 weeks) with two sessions a week?

Moreover, the intervention included TENS therapy (15 min), therapeutic massage (15 min) and either 3D-MCE or NDT. Since the aim of the study was to compare the effects of 3D-MCE and NDT, why didn’t the sessions include only 3D-MCE or NDT?

The duration of 3D-MCE or NDT in each session is short (15 min) compared to the duration usually used. Please, justify the choice.  

According to a recent meta-analysis in Journal of Clinical Medicine 2020; 9(9):3058, MCE with a frequency of three to five times per week with a duration of 20–30 min was found to be the most effective for individuals with NSLBP.

Please add the method of randomization. Add where that the intervention took place. Was the intervention in groups or individualized? Were the assessors the same at baseline and follow-up. Were they blinded to group allocation?

Were the exercises performed under the supervision of a physical therapist? Were they always the same during the intervention period? Did athletes have any feedback and corrections of the movements?

Was the intensity at which they exercised monitored throughout the sessions?

Please add the statistical analysis section at the end of the methodology

Limitations: Please add that recurrences of LBP were not examined.

Author Response

Dear. Reviewer 1

We thank you and the reviewers for your thoughtful suggestions and insights. The manuscript has benefited from these insightful suggestions. I look forward to working with you and the reviewers to move this manuscript closer to publication in Healthcare.

The manuscript has been rechecked, and the necessary changes have been made based on the reviewers’ suggestions. The responses to all comments have been prepared and attached herewith/given below.

We responded late due to personal and systemic issues during the revision process. I am genuinely sorry.

I appreciate your consideration. I look forward to hearing from you.

Sincerely,

Reviewer 2 Report

Comments and Suggestions for Authors

Dear Authors

The theme is interesting, the objectives are relevant and well defined, and the indicators seem appropriate to the objectives. However, I have some doubts about the study design used.

In this sense, my biggest concern is the following:

They opted for a quasi-experimental study, with 2 groups, considering the existence of homogeneity between the groups. However, only homogeneity in relation to age and BMI is demonstrated. Was homogeneity in relation to pain, PPT, ODI and LFA not considered?

Why were therapeutic massage and TENS therapy considered in intervention protocols? Could it not bias the result of some of the variables, such as pain?

Regarding the results:

Is the value of the Mean difference in pain in the 3D-MCE group (table 3), correct?

In the pressure pain threshold, the comparison between groups is reported to be non-significant on the right and significant on the left, in the erector spinae muscle, the data in figure 4 seem to suggest the opposite.

The paragraph in lines 233 and 234 is confusing “The between-group comparison indicated a significant difference in lumbar flexion ODI at the BPT (p > .05) (Figure 4, Table 5).”

Figure 4 refers to a post-hoc result. It is not clear in relation to what the post-hoc is done. It is suggested that in chapter 2. Material and methods, introduce a topic “statistical analysis” where the statistics used are explained.

In the discussion, the fact observed in the variation in mechanosensitivity between the left and right erector spinae should be better explained.

The same for what is mentioned between lines 295 and 302. Could methodological gaps be biasing these results?

Minor concerns:

Please explain the reason for excluding an element from the sample, line 81 and 82, and figure 1

Is the model identification of the TENS device, correct?

Please replace the acronym VAS, in table 3 and figure 4, with NRPS.

Author Response

Dear. Reviewer 2

We thank you and the reviewers for your thoughtful suggestions and insights. The manuscript has benefited from these insightful suggestions. I look forward to working with you and the reviewers to move this manuscript closer to publication in Healthcare.

The manuscript has been rechecked, and the necessary changes have been made based on the reviewers’ suggestions. The responses to all comments have been prepared and attached herewith/given below.

We responded late due to personal and systemic issues during the revision process. I am genuinely sorry.

I appreciate your consideration. I look forward to hearing from you.

Sincerely,

Round 2

Reviewer 1 Report

Comments and Suggestions for Authors

I am pleased with the present revised version. No further changes are required. 

Reviewer 2 Report

Comments and Suggestions for Authors

Dear authors

I appreciate the changes made to the paper, I believe they managed to clarify all the aspects highlighted.